# Supplement usage and doping attitudes in elite youth sports: The mediating role of dietary supplement acceptance

Jan Åge Kristensen[1]*, Tommy Haugen[2], Yngvar Ommundsen[1]

1 Department of Sport and Social Sciences, Child and Youth Sport Research Center, Norwegian School of Sport Sciences, Oslo, Norway, 2 Department of Sport Science and Physical Education, Faculty of Health and Sport, University of Agder, Kristiansand, Norway

* janak@nih.no

**Data Availability Statement:** All relevant data are within the paper and its Supporting Information files.

## Abstract

This study investigated whether dietary supplement acceptance mediated the relationship between supplement use and doping attitudes in youth sports. To this end, we employed a two-wave half-longitudinal design during a sports season (time point one [T1] to time point two [T2]). The sample consisted of 217 elite youth athletes (47% male; mean age = 16.98 years, standard deviation = 0.88) who competed in team sports (43%; N = 93; basketball, floorball, handball, and ice hockey) and individual sports (57%; N = 124; alpine skiing, biathlon, cross-country skiing, swimming, and tennis). The participants were recruited from eight Norwegian sports academy high schools that provide extracurricular, higher-level training and specialization for youth athletes. Results from structural equation modeling analysis indicated that dietary supplement acceptance (T2) mediated the positive relationship between supplement use (T1) and doping attitudes (T2) when accounting for prior levels of the mediator and the outcome variable. These findings suggest that when young athletes used dietary supplements at the start of the season to improve their performance, they were more likely to view the use of supplements as acceptable and to report more favorable attitudes toward doping at the end of the season six months later. For those seeking to prevent doping in youth sports, targeting athletes' views on the acceptable use of dietary supplements may be important.

## Introduction

For decades, athletes have experimented with various ways to improve performance and recover from injury. Some means and methods are prohibited and banned in sports, collectively called doping, whereas others are considered more or less acceptable. Examples of the latter are dietary supplements, which include various products that claim to be capable of enhancing athletes' short-term performances [1]. However, despite the World Anti-Doping Agency's (WADA) effort to develop, harmonize and coordinate anti-doping rules, athletes' use of doping is still frequently reported–also among young athletes. Due to their desire to enter professional sports and the higher stakes contingent upon their improved performance,

**Funding:** The author(s) received no specific funding for this work.

**Competing interests:** The authors have declared that no competing interests exist.

young athletes might be particularly prone to doping behaviors [2]. Castronova and Wagner [3] echo concerns about doping in sports and suggest that raising awareness of anti-doping measures among athletes themselves could significantly aid in the fight against doping. Rather than relying solely on centralized bodies, such as WADA, focusing on changing athletes' attitudes toward doping may be a more effective and successful strategy to combat doping by athletes [4, 5].

A large body of literature has proposed several factors that could affect attitudes toward doping [6, 7]. One of which is the use of dietary supplements, which has been suggested to elicit more permissive attitudes toward the use of banned substances [8]. Recent evidence also highlights the potential importance of athletes' judgments as to whether the use of dietary supplements is acceptable [9]. However, to the authors' knowledge, no research has investigated the relationship between supplement use, dietary supplement acceptance, and attitudes toward doping. Over the course of a competitive sport season, young athletes are faced with wear and tear caused by loss, injury, and stress, arguably making them more vulnerable of doping behaviors [2, 10]. Hence, the present study aimed to examine whether the acceptable use of dietary supplements mediates the supplement use and doping attitudes relationship during a sports season in youth sports.

## Supplement use in sports

The use of dietary supplements is widespread in sports and often refers to a range of products that are purposefully ingested to enhance performance, promote recovery, and prevent nutrient deficiencies [11]. Such products include vitamins, minerals, amino acids, enzymes, creatine, herbs, and other botanicals. In a meta-analysis, Knapik et al. [12] reported that approximately two-thirds (60%) of athletes use a dietary supplement and that the prevalence estimates may vary by elite level and sport. However, if used excessively or incorrectly, dietary supplements may also have the potential to impair athletic performance, damage health, and induce more positive attitudes toward doping [1, 13]. With respect to the latter, the notion is that the use of dietary supplements could influence athletes' tendency to feel comfortable taking substances to improve performance, which may elicit more favorable attitudes toward experimenting with stronger and prohibited performance-enhancing substances.

Previous studies have provided support for the positive relationship between supplement use and attitudes toward doping [8, 14, 15]. Barkoukis et al. [8] reported that users of dietary supplements held more positive attitudes toward doping compared with nonusers. This tendency was also supported by Backhouse et al. [14], who additionally found that supplement users were more in favor of competing in competitive situations that allowed doping compared with nonusers. Overall, these findings are in line with what is well-established in social cognition research [6], revealing that those who engage in the use of supplements tend to express more permissive attitudes toward the use of prohibited substances. However, such cognitive adaptation may not be fully explained by experimentation with dietary supplements. It has been suggested that judgments on the acceptable use of supplements also might play a part in doping behaviors [9].

## Judging the acceptability of using dietary supplements in sport

The mental processes by which athletes judge how acceptable it would be to use a dietary supplement are believed to be influenced by four different information cues. According to Fruchart et al. [9], the informational cues are reflecting (a) short-term success, (b) health consequences, (c) detectability–the likelihood of a positive doping test due to contamination of a banned substance, and (d) the perceived behavior of others (e.g., the coach, peers, or

competitors) and whether they favor using the supplement. Rather than seeking guidance from medical staff who do not appear to be principal advisors, athletes typically rely on coaches as their primary source of information and influence [16]. For example, it is possible to state that the higher the expectation for short-term success, the lesser the negative consequences for health and the more favorable the coaches' attitudes, the more likely are athletes to consider using a dietary supplement as acceptable.

Previous findings from Fruchart and colleagues [9] provide valuable insight into athletes' supplement acceptance. In general, adolescent athletes considered the use of a dietary supplement to be more acceptable compared with adult athletes. Accordingly, when the expectation for short-term success were high, the negative consequences for health were perceived as low, and the coach's attitude was favorable, adolescent athletes tended to view supplement use as acceptable. Conversely, adults judged the use of supplement as acceptable only when the negative consequences for health were low. Together, these findings highlight underlying psychological mechanisms which may regulate athletes' judgements on the acceptable use of supplements. The same mechanisms, however, is also believed to influence athletes' performance-enhancement by doping [14, 17].

## The present study

Research examining dietary supplements is rich, and sufficient evidence suggests that the use is widespread in sports and that the experimentation with supplements prompt more favorable attitudes toward prohibited substances [12, 14]. However, a relative dearth of research has investigated whether dietary supplement acceptance mediates the relationship between supplement use and doping attitudes during a sports season in youth sport. The present study aimed to address this knowledge gap to better understand supplement use as a potential forerunner to attitudes toward doping [18]. Providing a better understanding of athletes' position regarding their dietary supplement acceptance may also be a key to prevent supplement users from turning to doping [13]. Additionally, the present study also contributes to the fight against doping by raising awareness of anti-doping measures among athletes themselves, aiding centralized bodies in their preventative work. Based on the existing evidence, we hypothesized that athletes' experimentation with supplements as a mean to improve performance would positively predict their dietary supplement acceptance, which in turn, would positively predict more permissive attitudes toward doping (see Fig 1).

## Materials and methods

### Participants

Using a half-longitudinal design, a purposive sample of participants was recruited from eight Norwegian sports academy schools that provide extracurricular, high-level training and specialization for youth athletes. A total of 598 participants completed an identical package of questionnaires at the start of the season (time point one [T1]); among these athletes, 217 continued to provide data at the end of season (time point two [T2]), reducing the final sample size to 217 participants. Females comprised 53% (N = 115) of the sample and males 47% (N = 102). The participants ranged in age between 15 and 19 years (mean age = 16.98; standard deviation [SD] = 0.88) and competed in team sports (43%; N = 93; basketball, floorball, handball, and ice hockey) and individual sports (57%; N = 124; alpine skiing, biathlon, cross-country skiing, swimming, and tennis). The participants also reported having participated in organized training sessions in their sport for an average of 9.36 years (SD = 3.07), and most of the participants (86%) invested more than 11 hours in their sport per week. The participants were chosen because of their sports (e.g., similarities related to their physical characteristics

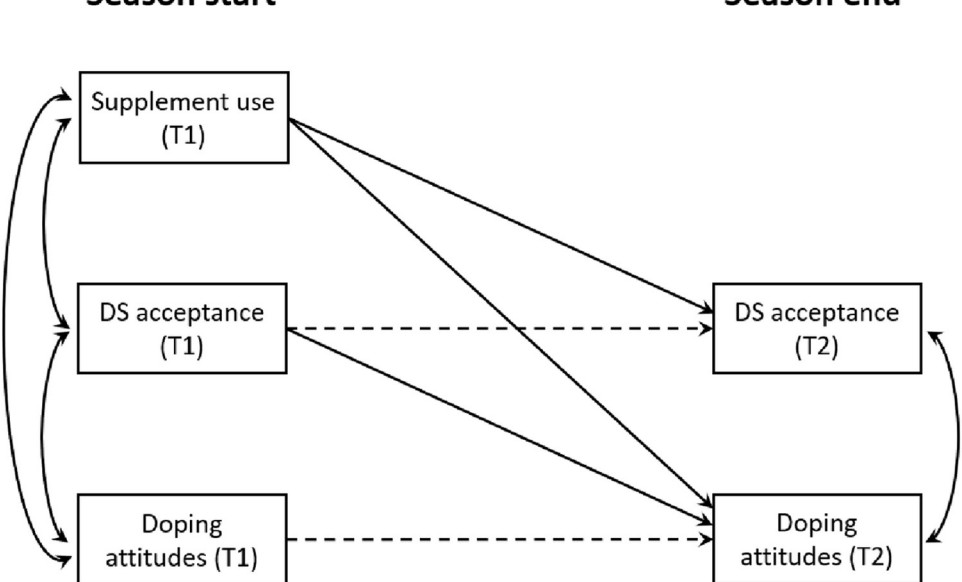

**Fig 1. The hypothesized mediation model for dietary supplement (DS) acceptance in the relationship between supplement use and doping attitudes.** Note: Time point one (T1) and time point two (T2) were separated by six months. The dashed lines represent the autoregressive paths.

and the period of the playing season) and their affiliation with the sports academies. The prestigious academies were regulated by competitive auditions and offered both acceleration and enrichment in the chosen sport. Hence, those participating in these schools could be regarded as being among the most ambitious and talented in their age group.

## Procedure

Before commencing the study, we sought ethical clearance from the University Ethics Committee of the first author's local institution. The national board of ethics and integrity in research, The Norwegian Centre of Research Data, approved the project prior to its commencement (reference no.: 571848). Recruiting a substantial number of participants in sports poses a significant challenge, especially when targeting adolescent athletes–a "hard-to-reach" population. Accessing the present study's sample required obtaining permission from the leaders of sports academies. After permissions were granted, all athletes associated with these academies were invited by the research team to participate in the study. In collaboration with sports academy representatives, an information sheet was forwarded to the participants and their parents/legal guardians. Written informed consent was obtained from all participants and parents of participants under 16 years of age, and their confidentiality was ensured. The data were collected using the SurveyXact digital survey tool [19], which stores the data on an encrypted server. In collaboration with the academies, the first author traveled to organize the data collection in separate activity groups, which helped monitor the data collection settings. Participants were also informed that consent could be withdrawn at any point without suffering any detrimental consequences. There were no withdrawals in the current study.

The first data collection began on September 9, 2020, coinciding with the start of the 2020–2021 sports season in Norway. A total of 668 questionnaires were distributed. However, seventy participants did not complete the survey, resulting in an 89.5% response rate (N = 598) at T1. We conducted the second data collection about six months later, on April 6, 2021, marking

the end of the 2020–2021 sports season. Out of the 598 participants at T1, 36.3% (N = 217) continued to provide data at T2. A series of independent *t*-tests with bootstrapping was conducted to investigate whether there were any differences between the distribution of observed variables between nondropouts and dropouts at T1 during the season start. The results indicated no significant differences. However, after running a bivariate correlation analysis using Cohen's evaluation of small ($r = 0.10$–$0.29$), medium ($r = 0.30$–$0.49$), and large effects ($r > 0.50$) to interpret effect size [20], some associations differed between the two groups. For those who remained in the study (i.e., nondropouts), the attitudes variable was small and medium associated with supplement use ($r = 0.24$, $p < 0.01$) and dietary supplement acceptance ($r = 0.41$, $p < 0.01$) at T1.

In contrast, for those who responded to only the first data collection at T1 (i.e., dropouts), the attitudes variable noted a nonsignificant association with supplement use ($r = 0.09$, $p = 0.10$) and a small association with dietary supplement acceptance ($r = 0.19$, $p < 0.01$) at T1. The observed z-value supported significant differences in associations between the two groups. Furthermore, it should be noted that the data collection took place during the COVID-19 lockdown in Norway. During the second data collection, several academies were placed under encompassing stay-at-home orders due to COVID-19 restrictions. This resulted in limited research opportunities and a reduced sample size, primarily due to organizational challenges related to the pandemic.

## Measures

All measurements were administered in Norwegian, following the translation-back-translation procedure from English [21]. We then tested a pilot version of the questionnaire on three youth athletes aged 16–17 years who gave feedback on the use of language and instructions before administering the survey. Only minor changes were made.

## Supplement use

In line with previous research [8], supplement use was measured with a single item ("How often do you use dietary supplements to improve your athletic performance?"). A definition derived from the 1994 Dietary Supplement Health and Education Act [22] including specific examples of supplements was provided and athletes were asked to recall their use over the last month (i.e., "Dietary supplement is defined as a product taken orally that contains a dietary ingredient intended to supplement the diet and may be found in many forms such as tablet, capsule, softgel, gelcap, liquid, powder, or bar. Examples of dietary supplements are vitamins, protein, creatine, and energy drinks"). The response options reflected the frequency of supplement use to improve athletic performance and were anchored on a five-point scale, with the options "never," "rarely," "sometimes," "frequently," and "very frequently."

## The acceptable use of dietary supplements

To assess athletes' views regarding the acceptable use of dietary supplements in sports, we used a hypothetical scenario adapted from Fruchart et al. [9], with some modifications made to enhance clarity and conciseness. The scenario corresponds to an actual sports situation and emphasize literature-based information cues on performance-enhancement in sports: (a) short-term success, (b) health consequences, (c) detectability, and (d) perceived attitudes of important others towards supplement use (e.g., the coach, peers, or competitors).

Jonas is a high-level athlete and a member of a renowned national club. He decided to absorb regular doses of PERFORM, a dietary supplement that significantly increases muscular mass and vital capacity. The use of this supplement is not banned, and it is totally

undetectable. In the long term, this product has no known negative effects on health. In the short term, it enhances performance and guarantees immediate success. Jonas's coach encouraged him to use this supplement.

After reading the scenario, participants were asked, "To what extent do you think that the use of PERFORM is acceptable". The participants indicated their responses on a seven-point scale (from 1 = not at all acceptable to 7 = completely acceptable), with a higher mean score indicating a stronger acceptance of using a dietary supplement. As such, the current scenario offers insight into athletes' acceptance of "clean" performance-enhancing substances. The use of scenarios has been shown to be valid in sports contexts [15, 23].

**Attitudes toward doping.**   Based on Ajzen's [24] recommendations, attitudes toward doping use were measured with the stem position "The use of doping substances to enhance my performance this season is . . ." followed by four semantic differential evaluative adjectives (bad/good, useless/useful, harmful/beneficial, and unethical/ethical) scored on a seven-point scale. A mean score was calculated, with higher scores reflecting more positive attitudes toward doping use. During the data collections, participants were also provided with WADA's [25] definition for doping, including examples of prohibited substances (i.e., "Doping is defined as the occurrence of one or more violations of the anti-doping rules in which athletes make use of substances and/or methods included on the Prohibited List in or out of competition. Examples of such prohibited substances are hormones, anabolic-androgenic steroids, and amphetamines"). The Omega coefficients [26] for the scale in this study were 0.78 and 0.76 in the first and second waves, respectively, indicating acceptable internal reliability [27]. This scale has been shown to be reliable and valid in several studies related to doping behaviors [17, 28].

## Data analysis

A combination of statistical methods was used to analyze the data. IBM SPSS Statistics version 28.0 (Armonk, NY: IBM Corp) was used to compute the descriptive statistics, reliability, and correlations. There were no outliers, and missing data on observed variables ranged from 0% to 3.7%. The result of Little's test of missing completely at random was nonsignificant (chi-square = 1.42, df = 13, $p$ = 1.00). Therefore, only the values pertaining to the participants with complete data were included in the analyses (listwise deletion). M*plus* version 8.5 [29] was used to examine the final mediation model. To determine model fit, we relied on common goodness-of-fit indices, including the chi-square test, comparative fit index (CFI), Tucker–Lewis's index (TLI), root mean square error of approximation (RMSEA), and standardized root mean square residual (SRMR). According to Geiser [30], a good fit is indicated by CFI and TLI values close to or greater than 0.90 and RMSEA and SRMR values less than 0.08. A priori power analysis for structural models was conducted [31], recommending a minimum of 140 participants to reach a power level of 0.8 to detect an anticipated effect size of 0.3 at an alpha level of 0.05, with 2 latent variables and 11 observed variables. However, in line with Hayes's [32] critique related to the estimation of the interaction of latent variables and the sample size of the current study, which could be regarded as low for latent variables modeling (N = 217), manifested variables were used in the structure model to ensure sufficient statistical power [20].

The half-longitudinal design was employed as the minimum required to test mediation and the most feasible, requiring only two measurement occasions [33]. As such, the present study's methodological approach stands out from previous studies on doping behavior, which mostly utilized a cross-sectional design and regression-based modeling [17, 34, 35]. Little [33] emphasizes the significant improvement in inferential power of the half-longitudinal model over cross-sectional mediation tests, as it permits control for prior levels and enables the assessment

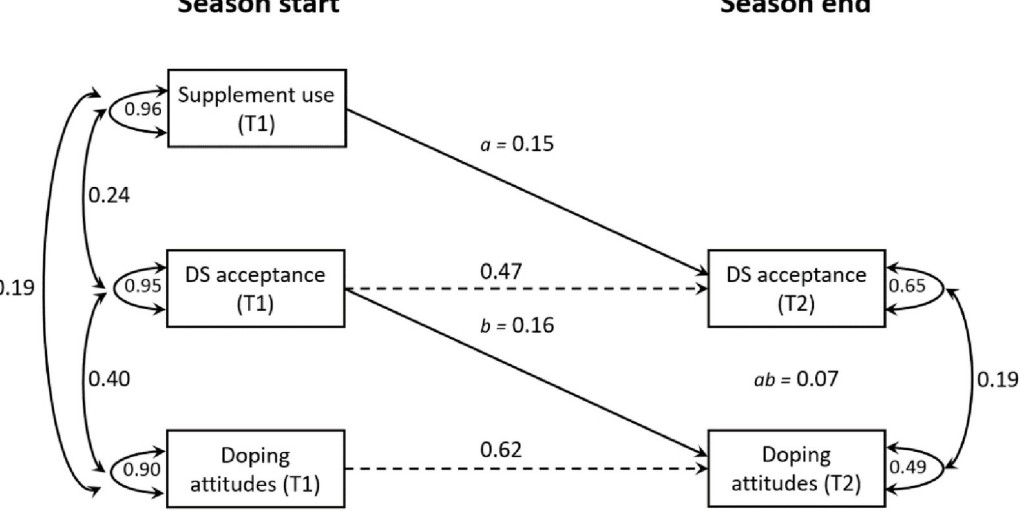

**Fig 2. Half-longitudinal mediation model for dietary supplement (DS) acceptance in the relationship between supplement use and doping attitudes.** Note: Time point one (T1) and time point two (T2) were separated by six months. The dashed lines represent the autoregressive paths. For ease of presentation, only significant (p < 0.05) standardized parameter estimates for the structural model are presented, and the covariate sex is not shown in the diagram.

of the influences on the change variance of the mediator and the outcome. We used structural equation modeling (SEM) and the maximum likelihood estimation with standard errors (ML) to test the half-longitudinal model (depicted in Fig 2). Guided by Little [33], we accounted for prior levels of the mediator and outcome variables to isolate the change variance. The primary paths of interest were the relationship between the predictor and the mediator, controlling for prior levels of the mediator (path *a*), and the relationship between the mediator and the outcome (path *b*), controlling for prior levels of the outcome. Assuming stationarity, the product of *ab* is an estimation of mediation, which was created for the mediator (i.e., dietary supplement acceptance) using the model constraint command in M*plus* [29]. The paths of interest were also explored using a bootstrapping procedure [36]. Bootstrapping generates an empirical representation of the sampling distribution (in the present study, 10,000 samples were drawn) and estimates the indirect effects in each resampled data set.

Compared with other mediation methods, bootstrapping has been found to be more robust to non-normal distribution and tends to have greater power to detect significant effects while allowing for the control of covariates [37]. Given the differences in doping behaviors among females and males [7], we included sex as a covariate. As participants were recruited from eight different sports academies, the possibility of clustering effects existed at the school level. Running multilevel SEM with a small number of clusters (< 30) may have led to biased estimates of the between-level standard errors and was therefore discarded [38]. Thus, to handle the shared variance between the academies, we employed a method that accounts for the nested data by adjusting standard errors and goodness-of-fit model testing [39]. This was done by specifying Type = Complex in M*plus*.

## Results

### Descriptive statistics and correlations

Table 1 presents descriptive statistics and correlations among all study variables for the total sample. As indicated by the mean values, at each wave, participants reported a low score for doping attitudes and a score below the midpoint for dietary supplement acceptance. The

**Table 1. Descriptive statistics and bivariate correlations for all variables across time (n = 209).**

| Variables | M (SD) | 1. | 2. | 3. | 4. | 5. | 6. |
|---|---|---|---|---|---|---|---|
| 1. Supplement use (T1) | 2.07 (1.03) | - | | | | | |
| 2. DS acceptance (T1) | 2.78 (2.01) | 0.28** | - | | | | |
| 3. DS acceptance (T2) | 3.10 (2.18) | 0.32** | 0.55** | - | | | |
| 4. Doping attitudes (T1) | 1.33 (0.68) | 0.24** | 0.43** | 0.35** | - | | |
| 5. Doping attitudes (T2) | 1.38 (0.68) | 0.24** | 0.44** | 0.42** | 0.71** | - | |
| 6. Sex | 1.53 (0.50) | –0.18** | –0.23** | –0.28** | –0.31** | –0.23** | - |

Note: Bootstrapped descriptive statistics and zero-order correlations generated from the sample of 217 athletes who contributed to the first and second data collections. The possible range of responses is 1–7 for all variables except supplement use (1–5). Sex was coded as 1 = males, 2 = females. DS acceptance, Dietary supplement acceptance; T1, time point 1/season start; T2, time point 2/season end

**p < 0.01 (two-tailed).

participants also reported a score below the midpoint for supplement use, indicating extensive use of supplements at season start. Prevalence estimates of the total sample showed that approximately two-thirds of participants (65%; N = 140; 69 females, 71 males) engaged in the use of supplements to improve performance. Of those, the majority reported the frequency of supplement use as rarely (51%; N = 72; 42 females, 30 males), sometimes (34%; N = 48; 23 females, 25 males), and frequently (11%; N = 15; 4 females, 11 males). Only a small number of participants indicated their use as very frequently (4%; N = 5; 5 males). Although most participants reported supplement use, some respondents (35%; N = 77; 46 females, 31 males) did not engage in such use to improve their performance (e.g., nonusers).

Furthermore, there was also a change in the respective mediator and outcome variables during the sports season. Paired sample $t$-tests revealed a statistically significant increase in the scores for dietary supplement acceptance from the season start to the season end (M$\Delta$ = 0.32 ± 0.18, $t$ (208) = –2.26, $p < 0.05$, 95% confidence interval [CI]: [–0.58, –0.04], Cohen's $d$ effect size [$d$] = 0.15). However, no statistically significant increase in doping attitudes was found (M$\Delta$ = 0.05 ± 0.01, $t$ (208) = –1.51, $p = 0.13$, 95% CI: [–0.13, 0.02], $d = 0.07$). Zero-order correlations revealed that doping attitudes at the end of the sports season was medium and small associated with dietary supplement acceptance ($r = 0.44$, $p < 0.01$) and self-reported supplement use ($r = 0.24$, $p < 0.01$) at the season start. In addition, a moderate correlation was found between doping attitudes and dietary supplement acceptance ($r = 0.42$, $p < 0.01$) at the end of the season. Together, these findings suggest that athletes who frequently used supplements at the season start were more likely to view their use as acceptable and, in turn, report more favorable attitudes toward doping at the season end.

## Testing the hypothesized mediation model

The main aim of the present study was to examine whether a change in dietary supplement acceptance mediates the effects of supplement use at season start on doping attitudes at the end of the sports season. The structural model generated for this purpose (see Fig 2) revealed an acceptable fit to the data $\chi^2(1) = 1.50$, $p = 0.22$, RMSEA = 0.05, 90% CI: [0.00, 0.20], CFI = 1, TLI = 0.98, and SRMR = 0.02. As shown in Fig 2, the use of supplements predicted the perceived acceptability of using dietary supplements at T2 ($\beta = 0.15$, 95% CI$_a$: 0.05, 0.25) while controlling for T1 levels of the mediator and the covariate sex ($\beta = -0.15$, 95% CI: –0.25, –0.05). Furthermore, a significant temporal link emerged between dietary supplement acceptance at T1 and doping attitudes at T2 ($\beta = 0.16$, 95% CI$_b$: 0.05, 0.28) while controlling for T1 levels of the outcome and sex ($\beta = 0.01$, 95% CI: –0.06, 0.08).

To probe the mediation, the product ab was created to explore how supplement use influences doping attitudes through dietary supplement acceptance. The product ab was statistically significant and different from zero (β = 0.07, 95% CI$_{ab}$: 0.01, 0.14), indicating that dietary supplement acceptance (i.e., the mediator) operate to facilitate the relationship between supplement use and doping attitudes. In addition, the autoregressive effects of dietary supplement acceptance (β = 0.47, 95% CI: 0.36, 0.58) and doping attitudes (β = 0.62, 95% CI: 0.51, 0.73) were sizable, suggesting reasonable stability of these constructs over time, with T1 measures being predictive of corresponding T2 levels. No significant direct path (i.e., c' path) between supplement use and doping attitudes at T2 were found (β = 0.03, 95% CI$_{c'}$: –0.07, 0.13). For more information related to the computer syntax [M*plus*] for the hypothesized mediation model, please see Supplementary material.

## Discussion

The aim of the present study was to test a half-longitudinal mediation sequence model in which the acceptable use of dietary supplements was hypothesized to mediate the relationship between supplement use and pro-doping attitudes. Our target group of athletes is of particular importance due to two main reasons. Firstly, they serve as a stepping stone to getting into professional sports. Secondly, they have a high prevalence of supplement use, which is associated with the strive for good performance. This combination makes them more prone to the likelihood of health-compromising behaviors, such as doping [2, 12].

The results showed a mediated sequence of relations between the predictor and the outcome variable: an indirect relationship between supplement use and doping attitudes through the acceptability to use dietary supplements. As illustrated in Fig 2, supplement use was related to dietary supplement acceptance, which in turn, was related to pro-doping attitudes. In statistical terms, the half-longitudinal model holds promise by demonstrating that the use of supplements predicts residual changes in dietary supplement acceptance when accounting for prior levels of the mediator, and dietary supplement acceptance predicts residual changes in doping attitudes when accounting for prior levels of the outcome [33]. These findings suggest that supplement use at the season start was related to higher acceptability to use dietary supplements, which in turn, was related to more favorable attitudes toward doping at the season end six months later.

Several studies have contributed to the doping literature and understanding of the supplement use and pro-doping attitudes relationship [8, 15]. Barkoukis et al. [8] investigated the complex relationship between supplement use and doping behaviors and found that supplement users become more favorable toward doping use, prior engaging with this behavior. Additionally, Hurst et al. [15] found that supplement use could lead athletes to beliefs about their effectiveness and favor more positive attitudes toward the use of prohibited substances–possibly due to perceived performance enhancement. Our findings add to the existing research by showing that dietary supplement acceptance mediate the relationship between supplement use and doping attitudes. It appears that experimentation with supplements aiming to improve performance may lead athletes to view dietary supplements as more acceptable, which in turn, might elicit more permissive attitudes toward experimenting with prohibited substances.

The results from the present study also underline the importance for young athletes to be able to discern between dietary supplements and prohibited substances [40]. Indeed, the latter substances are on a prohibited list and are thus unacceptable for use in and/or out of competition [25]. Complicating matters further, several substances introduced to the market as performance enhancers were at first treated as an innocent supplement before later being added to the prohibited list. Additionally, acceptable and prohibited substances share similar forms,

such as tablets or liquids, contributing to the difficulty in differentiation. Consequently, athletes lacking understanding in discerning between dietary supplements and doping may be more inclined to judge the use of prohibited substances as more positive once they are provided with various supplements and reassured that using them is acceptable.

As can be seen in the path diagram (see Fig 2), the autoregressive effects of dietary supplement acceptance and doping attitudes were sizable, indicating reasonable stability of the constructs during the season [41]. However, when inspecting the two autoregressive coefficients, dietary supplement acceptance appeared to be the construct that fluctuated more. While psychological constructs are often thought to be stable and enduring over time, some researchers favor the view that they are adaptive reactions to environmental demands [42, 43]. According to Schwarz [43], attitudes are context-specific judgments constructed from currently accessible information. Given the varied phases encountered during a sport season, the wear and tear caused by loss, injury, and stress may influence athletes' judgments to use supplements to a greater or lesser extent [10]. Consequently, in these situations, athletes may view their means as more acceptable and express more positive attitudes toward prohibited substances. Nevertheless, more knowledge of the temporal patterns of dietary supplement acceptance and doping attitudes in sports is needed [44].

Contrary to expectations, we did not observe a significant direct path between supplement use at season start and doping attitudes at season end. According to previous research [8], dietary supplement use would be expected to facilitate attitudes toward doping. However, prior research has also encountered challenges in establishing a direct path to doping attitudes [15]. When residualizing the mediator and outcome variable, there is a less room left for unexplained variance by the predictor (i.e., supplement use), making significant longitudinal links challenging to discern [45]. The sizable autoregressive coefficients herein signify a relatively small amount of residual change to be predicted, which may explain the nonsignificant direct path between supplement use and doping attitudes. Hence, the absence of a direct effect of supplement use on doping attitudes underline the significance of athletes' acceptance of dietary supplements as a potential explanatory mechanism for the link between supplement use and doping attitudes.

When studying the descriptive statistics in Table 1, mean levels indicated an increase in dietary supplement acceptance and doping attitudes from season start to season end; thus, the change was more substantial in dietary supplement acceptance. Notwithstanding the small effect sizes of dietary supplement acceptance and attitudes, as indicated by Cohen's $d$ [46], these findings suggest that participants viewed dietary supplements as more acceptable and judged the use of prohibited substances (i.e., attitudes toward doping) as more positive at the end than at the start of the season. However, due to the low mean in doping attitudes scores of the young athletes, their attitudes indicated vehement rejections of doping. The use of dietary supplements to increase the level of athletic performance in the present study sample appears to be in line with that of the general and collegiate population of similar age [12].

Overall, the findings from the present study may also bring important policy implications for anti-doping systems. Alongside enforcing regulations and punishments for violations of anti-doping rules, centralized bodies may also benefit from fostering a deeper internalization of the values of clean competition and preserving the true spirit of sports–such as fair competition and respect for rules and laws [47]. Providing young athletes narratives of "clean athletes" which entails success through dedication and determination on a level playing field without doping, could be a valuable approach to contribute to the global anti-doping program [48].

## Methodological issues and directions for future research

Even though interesting results emerged from the present study, the following limitations should be considered when making inferences. First, we made use of a half-longitudinal

design, which assumes stationarity between the mediator and outcome at additional time points and precludes directionality between variables based on a theoretical assumption [49]. It would be enlightening to replicate our findings using a full longitudinal mediation model with at least three time points to estimate a more rigorously estimated indirect pathway across the time spans [33]. Given the pace of change in the variables, using a sporting season would seem reasonable. However, to detect changes in doping attitudes, allowing longer intervals accompanied by a model that bears theoretical promise could also be prudent. Future research is also encouraged to broaden the age range of participants, include specific sports associated with a higher risk of doping, and explore heterogeneity between different groups. Second, due to the small number of clusters in the present study sample, we could not perform multilevel analyses. Therefore, future research is encouraged to approach the data analysis through multi-level SEM using a two-level rather than a traditional single-level model. Third, participants in the present study were recruited from specific sports academies with permission from key gatekeepers, such as the leaders of these schools. Consequently, a subtle selection bias among the athletes who opted to participate might have existed due to the school's support and inter-ests, potentially influencing athletes' decision-making to partake. Finally, the current study's data collections were performed during the COVID-19 lockdown in Norway, which dramati-cally reduced the sample size. Due to COVID-19 restrictions, athletes' competitive season were shortened, resulting in a reduced number of season's games. The potential impact of being in lockdown during the study could also have impacted athletes' supplement use, and potentially, their acceptability to use dietary supplements and their doping attitudes. Collecting data is never easy, and future researchers should be aware of the potential pitfalls when collecting and analyzing longitudinal data [50].

## Conclusion

This study aimed to investigate the relationship between supplement use and doping attitudes in youth sports. Previous research has mainly consisted of cross-sectional investigations, including beliefs about the effectiveness of dietary supplements and psychosocial trends among supplement users and nonusers [8, 15, 18]. However, to our knowledge, no previous research has examined the mediating role of dietary supplement acceptance in a prospective relation between supplement use and doping attitudes. Aiming to deal with this shortcoming, our results revealed that supplement use was related to the residual change in dietary supple-ment acceptance, which in turn predicted residual changes in doping attitudes. These findings indicated that when young athletes used dietary supplements at the start of the season to improve their performance, they were more likely to view the use of supplements as acceptable and to report more favorable attitudes toward doping at the end of the season six months later. The results from the current study add to the existing research and underline the potential risk of experimenting with supplements. Although athletes commonly use supplements to enhance performance, frequent usage may increase the risk of developing more positive attitudes toward doping by influencing athletes' acceptance of using dietary supplements. Therefore, targeting young athletes' views on the acceptable use of dietary supplements may be important for those seeking to prevent doping–especially for those athletes who already have incorpo-rated supplements into their daily routine. The findings also carry important policy implica-tions for global anti-doping systems, advocating a need to shift incentives more towards the athlete. In addition to combating doping through regulations and punishments, centralized bodies should also encourage the internalization of the "clean athlete" ideal among young athletes.

### Endnotes

The structural model was also analyzed using latent variables. In line with the reported priori power analysis, we performed the SEM analysis with 2 latent variables and 11 observed variables. However, the model did not fit the data well $\chi^2(44) = 150.88$, $p < 0.001$, RMSEA = 0.11, 90% CI [0.09, 0.13], CFI = 0.86, TLI = 0.79, and SRMR = 0.07. Exploring and comparing results for structural paths of the latent model and the manifested model showed that some paths estimates differed slightly. Importantly, the *b* path and *ab* product were found to be non-significant in the latent structural model. With these considerations in mind, as well as critique raised by Haye's [32] concerning the estimation of the interaction of latent variables and sample size of the present study, we proceeded with a structural model using manifested variables.

## Supporting information

**S1 File. The computer syntax [*Mplus*] for the hypothesized mediation model is available in the Supplementary material.**
(DOCX)

**S1 Data.**
(SAV)

## Author Contributions

**Conceptualization:** Jan Åge Kristensen.

**Data curation:** Jan Åge Kristensen.

**Formal analysis:** Jan Åge Kristensen, Tommy Haugen.

**Investigation:** Jan Åge Kristensen.

**Methodology:** Jan Åge Kristensen.

**Supervision:** Yngvar Ommundsen.

**Writing – original draft:** Jan Åge Kristensen.

**Writing – review & editing:** Tommy Haugen, Yngvar Ommundsen.

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
