## [Decision Letter · Decision Letter 0]

8 Nov 2023

PONE-D-23-31523Supplement usage and doping attitudes in elite youth sports: The mediating role of dietary supplement acceptancePLOS ONE

Dear Dr. Kristensen,

Thank you for submitting your manuscript to PLOS ONE. After careful consideration, we feel that it has merit but does not fully meet PLOS ONE’s publication criteria as it currently stands. Therefore, we invite you to submit a revised version of the manuscript that addresses the points raised during the review process.

The two referees who suggest revisions do not criticize the methodology and the results of the manuscript, but they suggest extensions that I also consider useful. However, I do not want to insist on these extensions.

What I do expect, however, are at least explanations (also in the manuscript itself) if you do not want to follow the extensions suggested by the reviewers. However, I would be pleased (and future readers would certainly be too) if you would constructively take up some of the extensions in the manuscript.

We look forward to receiving your revised manuscript.

Kind regards,

Gert G. Wagner, Professor

Academic Editor

PLOS ONE

Journal Requirements:

Additional Editor Comments:

I now have three referee reports that give three different recommendations: from "accept" to "major revisions". Based on my own reading, I suggest "minor revisions".

The two referees who suggest revisions do not criticize the methodology and the results of the manuscript, but they suggest extensions that I also consider useful. However, I do not want to insist on these extensions.

What I do expect, however, are at least explanations (also in the manuscript itself) if you do not want to follow the extensions suggested by the reviewers. However, I would be pleased (and future readers would certainly be too) if you would constructively take up some of the extensions in the manuscript.

Best regards

GGW

Reviewers' comments:

Reviewer's Responses to Questions

**Comments to the Author**

1. Is the manuscript technically sound, and do the data support the conclusions?

Reviewer #1: Yes

Reviewer #2: Yes

Reviewer #3: Partly

2. Has the statistical analysis been performed appropriately and rigorously? 

Reviewer #1: Yes

Reviewer #2: Yes

Reviewer #3: I Don't Know

3. Have the authors made all data underlying the findings in their manuscript fully available?

Reviewer #1: Yes

Reviewer #2: Yes

Reviewer #3: Yes

4. Is the manuscript presented in an intelligible fashion and written in standard English?

Reviewer #1: Yes

Reviewer #2: Yes

Reviewer #3: Yes

5. Review Comments to the Author

Reviewer #1: The paper, and the larger literature on doping, needs to pay more attention to a simple factual problem: What is the ontological difference, if any, between a dietary supplement and an impermissible drug? True, one is on a list of banned substances, the other is not: But that difference is administrative, not essential. Many banned substances were at first treated as an innocent supplement. Thus it is no wonder that athletes who are given a shot one day, and told that it is ok, are puzzled when they are given a shot another day, and are criticized for taking a banned drug. It's also not surprising that the current methods of doping control are complete and utter failures, as the authors report. It is not possible to control doping if the athletes do not support the system with their moral norms. Moral norms are what control rules in sports like golf (see Castronova and Wagner, 2009). Moral norms against doping will never emerge, so long as athletes are passive recipients/victims of bureaucratic decrees and enforcement of apparently arbitrary distinctions between "supplement" and "dope." This paper could improve an already substantial contribution by adding a paragraph emphasizing how difficult it is for athletes to understand the difference between a banned drug and a dietary supplement, since they both have the same form - shots and pills - and the the same intent - to improve sports performance. The authors could explain that this innocent ignorance explains the paper's finding: Young athletes who can't tell the difference between supplements and dope quite understandably become more accepting of dope once they are given a bunch of supplements are reassured that doing so is perfectly ok.

Reviewer #2: The selection of participants is influenced by coaches decision. It is difficult to get numerous participants in sports but this fact should have been decribed more detailed e.g. in discussion or intro.

Reviewer #3: This paper empirically investigates whether youth athletes' dietary supplement (DS) acceptance mediates the link between supplemental use and doping attitudes. The authors use a sample of 217 Norwegian youth athletes whom they observe and interview twice with a gap of six months. The authors run structural equation modeling and find evidence for a mediating role of DS.

I enjoyed reading the paper. It is well structured, well written, concise and I find the topic of great importance. I have the following comments that may help to improve the readability for an interdisciplinary journal audience.

1. To be honest, I am not very familiar with "half longitudinal mediation sequence models." As far as I can judge it, the methodology and analysis are sound and valid. However, it would be nice if the authors could put their methodology in context and summarize other statistical approaches used in the literature. In my field, the standard approach would be a multivariate regression models, either OLS or multinomial or logit/probit for dichotomized dependent variables. Panel data are clearly a plus, and then researchers would apply random or fixed effects models. At least as a robustness check for the appendix, wouldn't it make sense to run a multivariate model just using the first wave and then full sample (and them maybe a random or fixed effects model using both waves)?

2. If I understand it correctly, the authors basically use 0-7 Likert scales as outcome variables and use them unmodified assuming essentially linearity (although the mention "a mean score was calculated", p. 10/11). If that is correct, it would be beneficial if the authors could discuss the underlying statistical assumptions for this approach to be unbiased? Alternatively, would it be possible to dichotomize the variables in robustness checks?

3. Why just controlling for gender? Isn't it possible to control for other background variables that were collected (age, socio-demographics,...)? At minimum it should be feasible to control for the region (urban/rural or county or state,...) and the sport discipline?

4. Likewise, would it make sense to test for effect heterogeneity by discipline and/or region?

5. It would be helpful if the authors could put their study in a broader context of the current (vs. potentially alternative) doping systems. The discussion of the current system on page 11 is too late for my taste. Further, given the findings, could the authors elaborate on potential policy implications and expand the last sentence of the Conclusion?

6. The editor has published work that links socialization and doping attitudes that may be relevant for this paper. https://academic.oup.com/eurpub/article/26/3/520/2467413? Also, there is a literature on issues related to the current doping system that the authors could try to link to their findings (see #5), also to broaden the readership https://journals.sagepub.com/doi/abs/10.1177/0022002797041006002

6. PLOS authors have the option to publish the peer review history of their article (what does this mean?). If published, this will include your full peer review and any attached files.

Reviewer #1: **Yes: **Professor Edward Castronova

Reviewer #2: No

Reviewer #3: No

---

## [Author Response · Author response to Decision Letter 0]

20 Dec 2023

Replies to the academic editor and reviewer’s comments.

We want to thank the editor and reviewers for reviewing our manuscript. We truly appreciate the effort, time spent, and thorough job done. It pleases us to receive the opportunity to address further comments from the reviewers. 

The current response letter responds to each point subsequently raised by the reviewers. 

Reviewer #1:

The paper, and the larger literature on doping, needs to pay more attention to a simple factual problem: What is the ontological difference, if any, between a dietary supplement and an impermissible drug? True, one is on a list of banned substances, the other is not: But that difference is administrative, not essential. Many banned substances were at first treated as an innocent supplement. Thus, it is no wonder that athletes who are given a shot one day, and told that it is ok, are puzzled when they are given a shot another day and are criticized for taking a banned drug. It's also not surprising that the current methods of doping control are complete and utter failures, as the authors report. It is not possible to control doping if the athletes do not support the system with their moral norms. Moral norms are what control rules in sports like golf (see Castronova and Wagner, 2009). Moral norms against doping will never emerge, so long as athletes are passive recipients/victims of bureaucratic decrees and enforcement of apparently arbitrary distinctions between "supplement" and "dope." This paper could improve an already substantial contribution by adding a paragraph emphasizing how difficult it is for athletes to understand the difference between a banned drug and a dietary supplement, since they both have the same form - shots and pills - and the same intent - to improve sports performance. The authors could explain that this innocent ignorance explains the paper's finding: Young athletes who can't tell the difference between supplements and dope quite understandably become more accepting of dope once they are given a bunch of supplements are reassured that doing so is perfectly ok:

We appreciate the above suggestion and acknowledge that the manuscript would benefit from explaining how difficult it is for athletes to understand the difference between a banned drug and a dietary supplement. We agree and have added a paragraph in the Discussion. Therefore, please see lines 392–401, or read the following: “The results from the present study also underline the importance for young athletes to be able to discern between dietary supplements and prohibited substances (36). Indeed, the latter substances are on a prohibited list and are thus unacceptable for use in and/or out of competition (23). Complicating matters further, several substances introduced to the market as performance enhancers were at first treated as an innocent supplement before later being added to the prohibited list. Additionally, acceptable and prohibited substances share similar forms, such as tablets or liquids, contributing to the difficulty in differentiation. Consequently, athletes lacking understanding in discerning between dietary supplements and doping may be more inclined to judge the use of prohibited substances as more positive once they are provided with various supplements and reassured that using them is acceptable”. 

In addition, we are also grateful for the tip about the cited paper. In line with Castronova and Wagner (2009) reflections of how to combat doping, we have amended one sentence in the Introduction to better inform the reader about the interplay between centralized authorities and athletes themselves. Please see lines 55–60 or read the following: “Castronova and Wagner (3) echo concerns about doping in sports and suggest that raising awareness of anti-doping measures among athletes themselves could significantly aid in the fight against doping. Rather than relying solely on centralized bodies, such as WADA, focusing on changing athletes’ attitudes toward doping may be a more effective and successful strategy to combat doping by athletes (4, 5)”. 

Reviewer #2:

The selection of participants is influenced by coaches decision. It is difficult to get numerous participants in sports but this fact should have been decribed more detailed e.g. in discussion or intro:

We appreciate the critical comment and agree that the manuscript would benefit from a more detailed description of the recruitment procedure. Therefore, we have added the following to the manuscript. Please see lines 162–166, now running as follows: “Recruiting a substantial number of participants in sports poses a significant challenge, especially when targeting adolescent athletes – a “hard-to-reach” population. Accessing the present study’s sample required obtaining permission from the leaders of sports academies. After permissions were granted, all athletes associated with these academies were invited by the research team to participate in the study”. 

Following this comment, we would also like to stress that the leaders of the sports academies and their respective coaches were significant gatekeepers. To get access to the athletes, leaders from the academies needed to grant a permission. After receiving the permission from the academies, all athletes from the respective academy were invited to partake in the present study, without being exposed to a selection bias from their coach. However, we also acknowledge that as sports academies granted their permission and support, the athletes may have been exposed to a subtle selection bias with respect to partake in such a study like this which might be considered of great importance to the school leaders. Therefore, we have touched upon this issue in the Methodological issues and directions for future research section. Please see lines 459–463, or read the following: “Third, participants in the present study were recruited from specific sports academies with permission from key gatekeepers, such as the leaders of these schools. Consequently, a subtle selection bias among the athletes who opted to participate might have existed due to the school’s support and interests, potentially influencing athletes’ decision-making to partake”. 

Reviewer #3:

This paper empirically investigates whether youth athletes' dietary supplement (DS) acceptance mediates the link between supplemental use and doping attitudes. The authors use a sample of 217 Norwegian youth athletes whom they observe and interview twice with a gap of six months. The authors run structural equation modeling and find evidence for a mediating role of DS. 

I enjoyed reading the paper. It is well structured, well written, concise and I find the topic of great importance. I have the following comments that may help to improve the readability for an interdisciplinary journal audience. 

1. To be honest, I am not very familiar with "half longitudinal mediation sequence models." As far as I can judge it, the methodology and analysis are sound and valid. However, it would be nice if the authors could put their methodology in context and summarize other statistical approaches used in the literature. In my field, the standard approach would be a multivariate regression models, either OLS or multinomial or logit/probit for dichotomized dependent variables. Panel data are clearly a plus, and then researchers would apply random or fixed effects models. At least as a robustness check for the appendix, wouldn't it make sense to run a multivariate model just using the first wave and then full sample (and them maybe a random or fixed effects model using both waves)?

Thank you for the encouragement of placing the present study’s methodology in context of other statistical approaches. We agree and have added a short summary of previous research and our reasoning for why the half-longitudinal model is considered superior compared to cross-sectional investigations. Please see lines 267–273, now running as follows: “The half-longitudinal design was employed as the minimum required to test mediation and the most feasible, requiring only two measurement occasions (19). As such, the present study’s methodological approach stands out from previous studies on doping behavior, which mostly utilized a cross-sectional design and regression-based modeling (17, 34, 35). Little (19) emphasizes the significant improvement in inferential power of the half-longitudinal model over cross-sectional mediation tests, as it permits control for prior levels and enables the assessment of the influences on the change variance of the mediator and the outcome”. 

With respect to a robustness check using a fixed or random effects models, we do not see how this would bring more robustness to our analytical approach. Our reasoning for choosing the half-longitudinal model was guided by Little (2013: Longitudinal structural equation modeling. New York: the Guilford Press), noting that this model may be well suited for the task to test the mediation hypothesis. We touched upon this in the manuscript, on lines 286–288, and hope that our reasoning lines bring understanding to the reviewer of why we chose the half-longitudinal model as our analytical approach and thus forward the value of using it. In the manuscript, lines 286-288 runs as follows; Compared with other mediation methods, bootstrapping has been found to be more robust to non-normal distribution and tends to have greater power to detect significant effects while allowing for the control of covariates (37). 

2. If I understand it correctly, the authors basically use 0-7 Likert scales as outcome variables and use them unmodified assuming essentially linearity (although the mention "a mean score was calculated", p. 10/11). If that is correct, it would be beneficial if the authors could discuss the underlying statistical assumptions for this approach to be unbiased? Alternatively, would it be possible to dichotomize the variables in robustness checks? 

We appreciate this critical comment. We have included the underlying statistical assumptions on lines 447–449 in the Methodological issues and directions for future research section. The assumptions are related to stationarity between the measurement occasions and the nature of the mediation process. Additionally, these assumption was further tested. As such, the Results section include paired sample t-tests to examine the change in the mediator and outcome variables and the mediational process was investigated using the model constraint command in Mplus, which created the ab product. Thus, we do not feel that this needs to be explained further. 

Regarding the comment pertaining to dichotomization of the outcome variables in a robustness check, we would like to argue against doing so. On average, dichotomizing a continuous variable might lead to large standard errors, smaller effect sizes, less statistical power, and missed effects (MacCallum, R. C., Zhang, S., Preacher, K. J., & Rucker, D. D. (2002). On the practice of dichotomization of quantitative variables. Psychological Methods, 7, 19-40. DOI: https://psycnet.apa.org/doi/10.1037/1082-989X.7.1.19). Thus, by keeping the outcome variables as continuous variables, we are more likely to keep valuable information from the data and thus able to assess the change variance of the mediator and the outcome variable. Additionally, to be more robust to non-normal distribution, the present study included bootstrapping when investigating the indirect effect between supplement use and pro-doping attitudes (please see lines 282–285). 

3. Why just controlling for gender? Isn't it possible to control for other background variables that were collected (age, socio-demographics,...)? At minimum it should be feasible to control for the region (urban/rural or county or state,...) and the sport discipline?

We appreciate the suggestion to include other potential relevant demographic variables. In line with Ntoumanis and colleagues’ review on doping behaviors, gender has been linked as a well-established demographic regardless of age and sport discipline, and was therefore included in our study (Ntoumanis, Barkoukis, & Backhouse, 2014. Personal and psychosocial predictors of doping use in physical activity settings). Due to specific characteristics of the present study’s sample, such as (i) being ambitious, (ii) performing a high training load of hours each week, and (iii) partaking in sports academies, we argue that these athletes have similarities and thus are more homogenous. Thus, controlling for age and sports discipline seemed arbitrary. Particularly so due to the relatively small age range among the participants and the low number of participants within each sport discipline. 

With respect to the abovementioned, we also acknowledge that this could be considered a limitation of the present study. Hence, future studies should increase the age range and include specifics sports associated with a higher frequency of doping, while including a high number of participants when examining the heterogeneity between groups. We touched upon this on lines 454–456, and we have added the following to the manuscript: “Future research is also encouraged to broaden the age range of participants, include specific sports associated with a higher risk of doping, and explore heterogeneity between different groups”. 

4. Likewise, would it make sense to test for effect heterogeneity by discipline and/or region?

While we appreciate this question, we align with the answer presented above (see comment three), and decided not to test the effect of heterogeneity due to the small sample size represented in each sport disciplines. However, as participants were recruited from eight different sports academies, we controlled for the potential clustering effect from the school level in Mplus using the Type = Complex command, which handle the shared variance between the academies. 

5. It would be helpful if the authors could put their study in a broader context of the current (vs. potentially alternative) doping systems. The discussion of the current system on page 11 is too late for my taste. Further, given the findings, could the authors elaborate on potential policy implications and expand the last sentence of the Conclusion?

Thank you for this critical comment. We have touched upon this in our response to the first comment from the first reviewer, in which we added lines concerning the interplay between centralized authorities and athletes themselves when combating doping (e.g., in the context of prohibited means vs acceptable ones). Additionally, we also added lines addressing how the present study, with reference to one of the suggested articles (e.g., Castronova and Wagner), may contribute to aid in the fight against doping within the current doping system. Please see lines 55–60 in the revised manuscript or read the following: “Castronova and Wagner (3) echo concerns about doping in sports and suggest that raising awareness of anti-doping measures among athletes themselves could significantly aid in the fight against doping. Rather than relying solely on centralized bodies, such as WADA, focusing on changing athletes’ attitudes toward doping may be a more effective and successful strategy to combat doping by athletes (4, 5)”. 

In addition to the abovementioned, we also added lines in the ending part of the Introduction. Please see lines 127–129, or read the following: “Additionally, the present study also contributes to the fight against doping by raising awareness of anti-doping measures among athletes themselves, aiding centralized bodies in their preventative work”. 

With respect to the policy implications, we added lines in the Discussion section. Please see lines 437–444, or read the following: “Overall, the findings from the present study may also bring important policy implications for anti-doping systems. Alongside enforcing regulations and punishments for violations of anti-doping rules, centralized bodies may also benefit from fostering a deeper internalization of the values of clean competition and preserving the true spirit of sports–such as fair competition and respect for rules and laws (47). Providing young athletes narratives of “clean athletes” which entails success through dedication and determination on a level playing field without doping, could be a valuable approach to contribute to the global anti-doping program (48)”. 

In addition, we have added lines to the Conclusion concerning the potential implications for policy on lines 489–493, now running as follows: “The findings also carry important policy implications for global anti-doping systems, advocating a need to shift incentives more towards the athlete. In addition to combating doping through regulations and punishments, centralized bodies should also encourage the internalization of the “clean athlete” ideal among young athletes”. 

6. The editor has published work that links socialization and doping attitudes that may be relevant for this paper. https://academic.oup.com/eurpub/article/26/3/520/2467413? Also, there is a literature on issues related to the current doping system that the authors could try to link to their findings (see #5), also to broaden the readership https://journals.sagepub.com/doi/abs/10.1177/0022002797041006002:

Thank you for the cited articles. We do appreciate these sources, and we have linked them both to the present study’s Introduction and Discussion sections.

---

## [Editor Report · Decision Letter 1]

28 Dec 2023

Supplement usage and doping attitudes in elite youth sports: The mediating role of dietary supplement acceptance

PONE-D-23-31523R1

Dear Dr. Kristensen,

We’re pleased to inform you that your manuscript has been judged scientifically suitable for publication and will be formally accepted for publication once it meets all outstanding technical requirements.

Kind regards,

Nour Amin Elsahoryi, pHD

Academic Editor

PLOS ONE
---

## [Editor Report · Acceptance letter]

18 Jan 2024

PONE-D-23-31523R1 

PLOS ONE

Dear Dr. Kristensen, 

I'm pleased to inform you that your manuscript has been deemed suitable for publication in PLOS ONE. Congratulations! Your manuscript is now being handed over to our production team.

Kind regards, 

on behalf of

Dr. Nour Amin Elsahoryi 

Academic Editor

PLOS ONE